# Normal Values for the fT3/fT4 Ratio: Centile Charts (0–29 Years) and Their Application for the Differential Diagnosis of Children with Developmental Delay

**DOI:** 10.3390/ijms25168585

**Published:** 2024-08-06

**Authors:** Nina-Maria Wilpert, Roma Thamm, Michael Thamm, Jürgen Kratzsch, Dominik Seelow, Mandy Vogel, Heiko Krude, Markus Schuelke

**Affiliations:** 1NeuroCure Cluster of Excellence, Charité–Universitätsmedizin Berlin, Corporate Member of Freie Universität Berlin and Humboldt-Universität zu Berlin, D-10117 Berlin, Germany; nina-maria.wilpert@charite.de; 2Department of Neuropediatrics, Charité–Universitätsmedizin Berlin, Corporate Member of Freie Universität Berlin and Humboldt-Universität zu Berlin, D-13353 Berlin, Germany; 3BIH Charité Junior Clinician Scientist Program, Berlin Institute of Health at Charité–Universitätsmedizin Berlin, BIH Biomedical Innovation Academy, D-10117 Berlin, Germany; 4Department of Epidemiology and Health Monitoring, Robert Koch Institute, D-13353 Berlin, Germany; thammr@rki.de (R.T.); thammm@rki.de (M.T.); 5Hospital for Children and Adolescents, Center for Pediatric Research, University of Leipzig, D-04103 Leipzig, Germany; juergen.kratzsch@medizin.uni-leipzig.de (J.K.); mandy.vogel@medizin.uni-leipzig.de (M.V.); 6Berlin Institute of Health, Bioinformatics and Translational Genetics, D-10117 Berlin, Germany; dominik.seelow@bih-charite.de; 7Institute of Experimental Pediatric Endocrinology, Charité–Universitätsmedizin Berlin, Corporate Member of Freie Universität Berlin and Humboldt-Universität zu Berlin, D-13353 Berlin, Germany

**Keywords:** thyroid hormone, TSH, reference values, fT3/fT4 ratio, peripheral thyroid hormone resistance, MCT8 deficiency, *THRA* mutations, *SECISBP2* mutations, *SLC16A2* mutations

## Abstract

Primary congenital hypothyroidism is easily diagnosed on the basis of elevated plasma levels of thyroid-stimulating hormone (TSH). In contrast, in the rare disorders of thyroid hormone resistance, TSH and, in mild cases, also thyroid hormone levels are within the normal range. Thyroid hormone resistance is caused by defects in hormone metabolism, transport, or receptor activation and can have the same serious consequences for child development as congenital hypothyroidism. A total of n = 23,522 data points from a large cohort of children and young adults were used to generate normal values and sex-specific percentiles for the ratio of free triiodothyronine (T3) to free thyroxine (T4), the fT3/fT4 ratio. The aim was to determine whether individuals with developmental delay and genetically confirmed thyroid hormone resistance, carrying defects in Monocarboxylate Transporter 8 (MCT8), Thyroid Hormone Receptor alpha (THRα), and Selenocysteine Insertion Sequence-Binding Protein 2 (SECISBP2), had abnormal fT3/fT4 ratios. Indeed, we were able to demonstrate a clear separation of patient values for the fT3/fT4 ratio from normal and pathological controls (e.g., children with severe cerebral palsy). We therefore recommend using the fT3/fT4 ratio as a readily available screening parameter in children with developmental delay for the identification of thyroid hormone resistance syndromes. The fT3/fT4 ratio can be easily plotted on centile charts using our free online tool, which accepts various SI and non-SI units for fT3, fT4, and TSH.

## 1. Introduction

Normal development in children depends on adequate thyroid function. This central role of thyroid hormones is best documented in neonates affected by congenital hypothyroidism, either due to the absence of the thyroid gland or due to defects in thyroid hormone synthesis. These children suffer from severe motor, language, cognitive, and somatic developmental delays [1]. If detected early enough through newborn screening and treated with high doses of thyroid hormone, these newborns have a normal neurodevelopmental and somatic outcome [2].

The regulation of thyroid function depends on a negative feedback loop involving the secretion of thyroid-stimulating hormone (TSH) by the pituitary gland. Consequently, children with congenital hypothyroidism and reduced thyroid hormone production are diagnosed on the basis of elevated plasma TSH levels [3]. However, in addition to congenital hypothyroidism associated with elevated TSH levels, there are several other rare conditions in which neurodevelopment is affected by a defect within the thyroid system despite normal TSH levels. In these cases, the thyroid gland produces thyroid hormone normally, but subsequent metabolism, transport, or receptor function may be impaired (Figure 1) [4].

### 1.1. Thyroid Hormone Metabolism

The primary hormone secreted by the thyroid gland is thyroxine (T4), a prohormone that must be metabolized to the active hormone, triiodothyronine (T3), by deiodination by two enzymes, deiodinases 1 and 2 (DIO1 and DIO2), which are members of the selenoprotein family [5]. T3 can also be synthesized directly on tyrosine residues of thyroglobulin in the thyroid gland. The majority of plasma T3 is produced in the liver and kidneys, where T4 is taken up and deiodinated via DIO1 to T3, which is then re-secreted into the circulation. Deiodination of T4 can already occur in the thyroid gland. Thus, under physiological conditions, direct T3 output from the thyroid accounts for up to 20% of plasma T3 (Figure 1) [6]. In addition, T4 can also be deiodinated to active T3 in the target cells themselves, where T4 is locally activated by DIO2 [5].

A specific rare disease affecting T4 metabolism has been identified: SECISBP2 deficiency. SECISBP2 is a common cofactor for the synthesis of selenoenzymes, including deiodinases [7]. Mutations in the *SECISBP2* gene reduce the deiodination of T4 to T3, resulting in mildly elevated plasma T4 and low-normal T3, while TSH is in the normal range. The clinical phenotype of patients partially mimics hypothyroidism, including developmental delay and cognitive dysfunction [8]. Most likely, the clinical symptoms result from reduced target cell deiodination of T4 and/or lower available plasma T3 levels. However, it is possible that dysfunction of other selenoenzymes may also contribute to the phenotype of patients.

### 1.2. Thyroid Hormone Transport

In addition to activation by deiodination, thyroid hormones must be transported to the liver and kidney for metabolism to target cells, such as neurons, to exert their physiological function. This transport is complex, and to date, we know of 15 transport proteins for the transport of T3 or T4. Different target cells show specific patterns of transporter expression [9]. Thus, a defect in a particular transporter protein will result in a specific phenotype in particular target cells depending on the particular expression of their transporter proteins. For example, it appears that the absence of the MCT8 transporter, the best-known and most studied thyroid hormone transporter to date, cannot be compensated for in the CNS. This leads to a rare disease with severe developmental delay and a movement disorder called Allan–Herndon–Dudley Syndrome (AHDS) [10,11]. In contrast, other organs such as the brain, heart, and liver are not hypothyroid in AHDS patients, most likely due to the expression of other thyroid hormone transporters that compensate for the MCT8 deficiency. The MCT8 deficiency leads to an imbalance in T4 metabolism, resulting in elevated T3 and rather low T4 plasma concentrations. Thus, in addition to a hypothyroid brain, the elevated plasma T3 can cause a state of peripheral hyperthyroidism of the liver and heart, although the exact molecular pathophysiology of these plasma changes is unknown.

### 1.3. Thyroid Hormone Receptors

Within the target cells, T3 acts as a transcriptional regulator because the thyroid hormone receptor (THR) is a transcription factor that can be either activated or inhibited by the ligand T3, depending on the cofactors involved [12]. Two different THRs are known, THRα and THRβ, which are encoded by two different genes, *THRA* and *THRB,* respectively. While *THRA* is expressed in many tissues and mainly in the CNS, *THRB* is preferentially expressed in the pituitary gland and is thus involved in the negative feedback control of the plasma thyroid hormone concentration via TSH. Patients with a THRβ defect have impaired thyroid hormone sensing by the pituitary, resulting in elevated TSH and consequently elevated T4 and T3, by which the diagnosis can be easily made. Clinical symptoms of these children may result from overstimulation of the unaffected THRα by the elevated plasma hormones, leading to hyperthyroid symptoms [13,14]. In contrast, a genetic defect in *THRA* does not result in elevated TSH because THRα is not involved in the feedback loop; instead, a local defect in THRα function within the target cells results in a local hypothyroid state that causes clinical symptoms similar to those of congenital hypothyroidism, such as developmental delay and cognitive dysfunction [15,16]. In addition, a defect in THRα function in metabolizing organs such as the liver and kidney results in a specific pattern of mildly elevated or high-normal T3 and, conversely, of mildly decreased or low-normal T4 [17,18]; again, the exact mechanism of these plasma hormone changes is not yet known.

Thus, while congenital hypothyroidism can be easily diagnosed by an elevated TSH, which is already established in newborn-screening programs [1], the three other genetic defects affecting thyroid metabolism, thyroid hormone transport, and thyroid hormone receptor function cannot be diagnosed on the basis of TSH. In severe cases of these rare diseases, measurement of T4 and T3 would allow for the detection of elevated T4 in SECISBP2 deficiency and elevated T3 in MCT8 deficiency, but in many cases, especially in THRα deficiency and mild cases of MCT8 and SECISBP2 deficiency, T4 and T3 levels are within the normal range and the diagnosis may be missed. However, in all three disorders, the plasma hormone levels are reciprocal abnormal, so the ratio of T3 to T4 tends to be more abnormal than it is with either single hormone level; thus, the T3/T4 ratio may be of diagnostic value.

To date, changes in the T3/T4 ratio have only been described in a few cases and in small cohorts, e.g., in patients with hyper- and hypothyroidism, in children with MCT8 deficiency [19], in children with congenital hypothyroidism treated with L-thyroxine, and in relation to iodine status in children [20,21,22], but no normal values have been reported. However, the T3/T4 ratio has not been evaluated as a tool in the clinical diagnostic process, because of the lack of age-dependent reference data for the T3/T4 ratio in the form of pediatric centile charts. Therefore, we set out to establish reference data for the T3/T4 ratio to allow early diagnosis of children affected by motor, language, and cognitive developmental delay, as well as muscular hypotonia due to *SECISBP2*, *SLC16A2*, or *THRA* gene defects. In addition, we tested the discriminatory power of these centile charts to single out the above-mentioned genetic defects using previously published and our own patient hormone levels (Appendix A).

**Figure 1 ijms-25-08585-f001:**
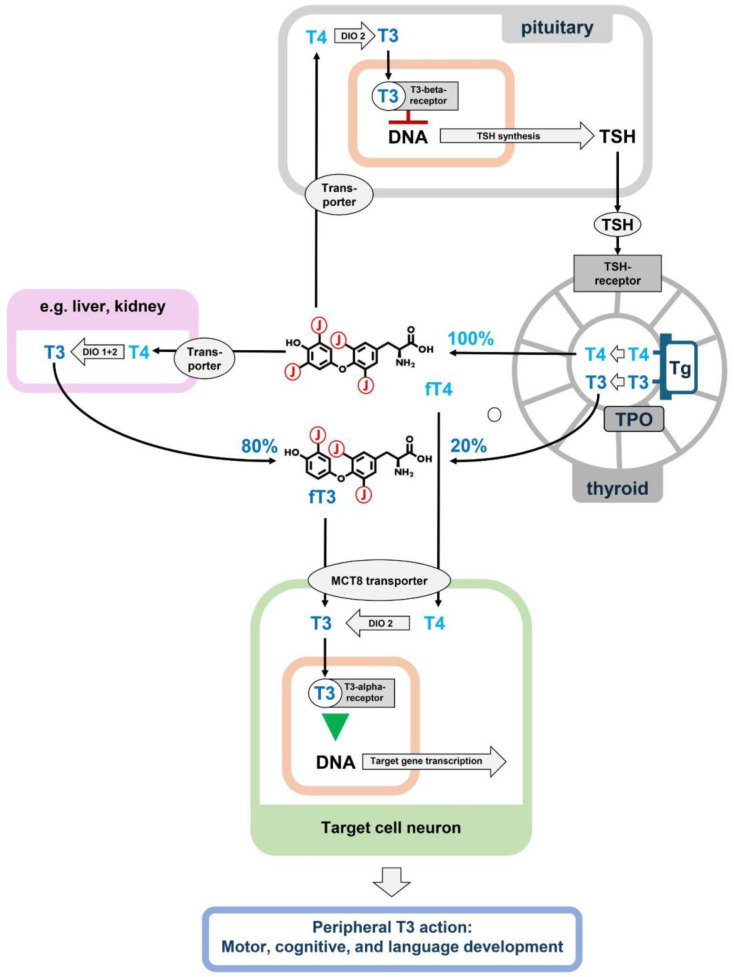
Thyroid hormone production, metabolism, and action on target cells. T4 is produced exclusively in the thyroid gland by iodination of thyroglobulin (Tg) by thyroid peroxidase (TPO), which is the sole source of plasma T4. Plasma T4 can be further deiodinated in peripheral tissues by deiodinases 1 and 2 (DIO 1 + 2) to T3, which is then red-secreted into the circulation and accounts for 80% of plasma T3, while 20% of plasma T3 is produced primarily in the thyroid gland. The actual ratio of plasma T3 to T4 is, therefore, the result of variable primary production in the thyroid, tightly regulated peripheral metabolism of T4 to T3, and its re-secretion. In the target cells, only the active hormone T3 binds the thyroid hormone receptors alpha and beta. To exert this final T3 effect, T3 can be transported directly from the plasma to the target cell, or it can be produced locally in the target cell from T4 by deiodinase 2. In the pituitary, the T3 effect leads to downregulation of TSH production, which is the key step in the negative thyroid–pituitary feedback loop that keeps thyroid hormone levels stable in the plasma [5,18].

For this purpose, we chose free, non protein-bound thyroid hormone levels because (i) these are the most commonly used parameters in clinical practice and (ii) fT3 and fT4 levels were available from two large German child health surveys, the “LIFE” and “KiGGS” studies [23,24]. We show here that the percentiles for the fT3/fT4 ratio show an age- and sex-specific normal range and that the individual fT3/fT4 ratio values of patients affected by a *SECISBP2*, *SLC16A2*, or *THRA* gene defect are outside the normal range. Therefore, we suggest the use of these fT3/fT4 ratio percentiles as a simple tool for clinical differential diagnosis in patients with motor and cognitive developmental delays.

## 2. Results

To establish the fT3/fT4 ratio as a clinically applicable serum/plasma-derived parameter for the differential diagnosis of children with developmental delays, we generated fT3/fT4 percentiles based on fT3 and fT4 measurements obtained in two large German cohorts of children, adolescents, and young adults. The combined data set represents n = 23,522 data points from individuals ranging in age from birth to 29 years (Appendix A).

### 2.1. Centile Charts for TSH, fT3, fT4, and fT3/fT4 Ratio

Consistent with previously published data, the TSH percentiles decreased steadily in both sexes throughout the 0–29-year age range (Figure 2a). Consistently with the decrease in TSH levels, and also consistently with published data [22,24], the fT3 and fT4 curves of our two cohorts showed a downward slope with increasing age (Figure 2b,c). In both cohorts, thyroid hormone levels were highest in the neonatal period and then declined, reaching a plateau at around 22 years of age. Interestingly, fT3 levels at birth (50th percentile, 7.05 pmol/L) declined by 33% to the lowest level in adulthood (50th percentile, 4.7 pmol/L), whereas fT4 levels declined by only 25% from the highest level (50th percentile, 16.5 pmol/L) to the lowest level (50th percentile, 12.5 pmol/L). Around the age of pubertal onset, the decline in both hormones was slowed, with a stronger effect on fT3 than on fT4, and the most pronounced effect was on fT3 in boys, who remained on a plateau between 8 and 14 years of age.

Given the parallel decline in fT3 and fT4 levels, one would expect a more even age distribution of the fT3/fT4 ratio. However, we saw a clear age- and sex-dependent course (Figure 2d), with higher values at birth and lower values in adulthood. Interestingly, the fT3/fT4 ratio peaked at 11.2 years in girls and 13.3 years in boys, roughly corresponding to the difference in the onset of puberty in the two sexes. The overall range of the fT3/fT4 ratio from the highest 97th percentile value in 13.3-year-old boys to the lowest 3rd percentile value in 17-year-old girls was 0.59 to 0.27. Thus, to evaluate the individual fT3/fT4 ratio in the context of clinical diagnosis, it is necessary to use percentiles that reflect this wide range of normal values at different ages and in different sexes.

### 2.2. fT3/fT4-Ratios of Different Patient Cohorts

Next, we tested the percentiles for the fT3/fT4 ratio for their discriminatory power between pathologies that are either thyroid-related (e.g., in patients with mutations in *THRA*, *SLC16A2*, or *SECISBP2*) or not (e.g., in patients with cerebral palsy).

We first evaluated data from patients with *THRA* mutations extracted from the literature [14,16,25,26,27,28,29]. While the individual fT3 and fT4 levels were mostly within the normal range, the fT3/fT4 ratio was above the 97th percentile in most patients. Only in n = 2 specific patients with a mutation in the most C-terminal amino acid residue 403 of THRα was the fT3/fT4 ratio lower, but still at or above the 90th percentile (Figure 3).

In patients with MCT8 deficiency and proven *SLC16A2* mutations, the fT3/fT4 ratios were clearly above the 97th percentile (Figure 4a). This was partly expected, as patients with MCT8 deficiency tend to have high fT3 plasma concentrations. However, this increase in the fT3/fT4 ratio was also observed in two very mildly affected patients in whom AHDS would not have been the first differential diagnosis and in whom an *SLC16A2* mutation could easily have been overlooked due to their mild phenotype and partially normal fT3 levels. Interestingly, the repeated measurements of the mildly affected patients all clustered around a similar fT3/fT4 ratio and were only slightly above the 97th percentile (Figure 4a). To better illustrate the wide range of neurodevelopmental phenotypes observed in patients with MCT8 deficiency, all of which been shown to have an elevated fT3/fT4 ratio, we have included a detailed description of the clinical phenotypes, accompanied by video sequences for a patient with a severe phenotype (Patient 1), a patient with a milder phenotype (Patient 2), and a patient with a very mild manifestation of the disease (Patient 3) (see Section 4.1.5).

As a proof of concept, we also discovered elevated fT3/fT4 ratios in two female patients with MCT8 deficiency as well as heterozygous *SLC16A2* mutations with skewed X-inactivation. Both patients were mildly affected, mainly by delays in language development. In one of these females, treatment with levothyroxine in combination with the deiodinase inhibitor 6-n-propyl-2 thiouracil (PTU) [30] normalized the fT3/fT4 ratio, possibly due to the combination of inhibition of T4-to-T3 deiodination and increase in T4 levels by supplementation (Figure 4b).

As expected from the previously described rather elevated T4 and low-normal T3 levels in patients with SECISBP2 defects [31,32,33,34], we found that the fT3/fT4 ratios of these particular patients were well below the 3rd percentile (Figure 5).

Finally, as a negative control, we tested the fT3/fT4 ratio in a cohort of n = 12 children between 1.3 and 16.7 years of age with severe cerebral palsy (CP) due to perinatal brain damage who had come to our outpatient department for treatment with botulinum toxin injections. Patients in this control cohort were also affected by severe symptoms such as spasticity, dystonia, and weight loss in addition to developmental delay, which partially overlap with the symptoms seen in patients with MCT8 deficiency. All values in these children were well within the normal ranges, regardless of age or sex (Figure 6).

## 3. Discussion

Primary congenital hypothyroidism is easily diagnosed on the basis of elevated TSH and low T3 and T4 levels [3]. Since the discovery of the various forms of thyroid hormone resistance, it has been shown that these rare disorders often have normal TSH, T3, and T4 levels and are therefore likely to be massively underdiagnosed [35]. Patients with these disorders have developmental and cognitive deficits that can be as severe as those associated with congenital hypothyroidism.

Despite normal absolute T3 and T4 levels, the ratio of the two hormones appears to be altered in these thyroid hormone-resistant disorders. However, the lack of normal age- and sex-specific values for the fT3/fT4 ratio has hampered the establishment of this serum/plasma parameter for screening and differential diagnosis of developmental delay.

Here, we report the successful establishment of age- and sex-specific normal values for the fT3/fT4 ratio. Based on two large German population-based cohorts from the “KiGGS” and “LIFE” studies, more than 23,000 data sets were available to derive the percentile ranges for fT3, fT4, TSH, and the fT3/fT4 ratio. The large data set of fT3/fT4 ratios is normally distributed. From birth, the mean values decrease until they show a temporary upward trend around the onset of puberty, with a significant difference between boys and girls. Therefore, the use of age- and sex-specific percentiles is mandatory for evaluation and clinical judgment.

When investigating the use of the newly generated fT3/fT4 percentiles for the diagnosis of three thyroid hormone resistance disorders, e.g., THRα, MCT8, and SECISBP2 deficiency, we saw a clear separation of the individual patients’ fT3/fT4 values from the normal percentile ranges. Even patients with THRα- and MCT8 deficiency with completely normal absolute T3 and T4 levels were found to have elevated fT3/fT4 ratios above the 97th percentile. Only two patients with the most C-terminal mutations in residue 403 of the *THRA* gene were found to have lower fT3/fT4 ratios, which may indicate a specific genotype–phenotype correlation regarding the effect of THRα deficiency on T4 metabolism.

It seems reasonable to use the fT3/fT4 ratio as another parameter for the differential diagnosis of children with developmental delays but normal TSH levels. Despite the expectation that an increasing number of these children will be diagnosed via the thorough and early implementation of whole-exome sequencing (WES) [36,37], the fT3/fT4 ratio will help to interpret variants of unknown significance (VUS) in *SLC16A2* or *THRA*, and may direct the search towards non-coding areas of the genome that represent around 60% of cases [38].

This is especially true due to the therapeutic options that are available, such as the treatment of patients with *THRA* mutations with L-thyroxine [39], and that will be developed in the near future for these thyroid hormone resistance diseases. Although the full spectrum of symptoms cannot entirely be compensated, treatment of THRα and SECISBP2 defects with additional L-thyroxine has been shown to improve some clinical features such as growth retardation [32,40,41]. Based on the experience with congenital hypothyroidism, treatment needs to be established very early in development to improve cognitive and motor outcomes. Our results, with the finding of an altered fT3/fT4 ratio already in very young children (the youngest tested child was 0.9 years old), open the possibility that the fT3/fT4 ratio could be a tool to detect these diseases in the neonatal period, probably as a second-tier test after the initial genetic newborn screening [42].

Our centile charts of the fT3/fT4 ratio are based on SI units (pmol/L). Unfortunately, there is a wide range of units used for hormone measurements, which can be confusing and can hinder the correct calculation of the fT3/fT4 ratio for an individual patient. Therefore, we provide a web page, http://www.thyroid-hormone-ratio.org (accessed on 29 June 2024), which allows the hormone measurements to be entered in a wide range of units and provides a printout that visualizes the individual’s TSH, fT3, and fT4 measurements, as well as the fT3/fT4 ratio within the age- and sex-adjusted centile chart.

The molecular pathophysiology of the elevated fT3/fT4 ratio is currently unknown. Our charts clearly suggest an age- and sex-dependent regulation. The regulator could be located at the area of primary hormone production, e.g., in the thyroid gland, or could be the consequence of different deiodination rates in those organs that contribute to secondary T3 production from T4, such as the liver, kidney, and muscle [5,18]. A recent meta-study of genome-wide associations described 13 gene loci contributing to the plasma fT3/fT4 ratio, including the two deiodinase gene loci *DIO1* and *DIO2*. However, the contributions of all these loci together explain only a few percentage points of the variance [43]. In addition, the (patho)physiological relevance of the fT3/fT4 ratio remains unclear because organ cells that are targets of thyroid hormone are able to locally generate their own T3 through intracellular deiodination of T4 (Figure 1).

## 4. Materials and Methods

### 4.1. Description of the Study Samples

#### 4.1.1. KiGSS Baseline Study (Robert Koch Institute, Berlin)

The KiGGS baseline study (German Health Interview and Examination Survey for Children and Adolescents) was the first nationwide study on the health of children aged 0–17 years and was conducted by the Robert Koch Institute (RKI) of Berlin between 2003 and 2006 [22]. It included n = 17,641 children from 167 communities in Germany. The study collected representative data on physical and mental health (including thyroid health), health behaviors, and determinants based on physical and laboratory investigations and interviews. The data were adjusted to exclude individuals with potential thyroid disease, e.g., individuals with TSH plasma levels above 10 mU/L or medications affecting thyroid function. Laboratory tests for thyroid health were performed in n = 12,836 individuals aged 3–17 years, including plasma levels for TSH, fT3, and fT4, as well as thyroid volume by ultrasound and the urinary iodine/creatinine ratio as a marker for iodine supply [22,44,45,46].

#### 4.1.2. KiGGS Wave 2 Study (Robert Koch Institute, Berlin)

The second follow-up study (KiGGS Wave 2) was again conducted as a health examination and interview survey between 2014 and 2017, with study participants now aged 10–31 years and living in the same recruitment areas as in the baseline study (n = 10,853) [22]. Many of them had also participated in the first follow-up (KiGGS Wave 1). KiGGS Wave 2 also included a new cross-sectional interview and examination survey. For n = 6465 participants, further physical and laboratory investigations were carried out; for n = 2971 study participants, we have laboratory values reflecting thyroid health (TSH, fT3, fT4). For n = 2643 study participants, we have the corresponding laboratory values for the age group between 18 and 29 years.

#### 4.1.3. LIFE Child Study (University of Leipzig)

The longitudinal epidemiological LIFE Child Study is part of the LIFE Research Center for Civilization Diseases and has been collecting developmental data on infants, children, and adolescents since 2011, primarily focusing on the so-called civilization diseases [47], with a special focus on thyroid health [24,48]. For our centile chart calculation, we used TSH, fT3, and fT4 values from children aged 0–6 years using n = 5413 data points from n = 2287 individuals. Within this period, children were sampled between 1 and 8 times (mean = 2.3, SD = 1.51).

#### 4.1.4. Measurement of Thyroid Hormone Values

LIFE Child Study: After an overnight fast, venous blood was collected in the morning. TSH, fT3, and fT4 values were measured by electrochemiluminescence assay using a Cobas 601 or 801 module (Roche Diagnostics, Mannheim, Germany). The mean inter-assay coefficient of variation for the three measured biomarkers ranged from 2.25% to 3.11%; the mean deviation from the target value ranged from 3.33% to 4.82% [24].

KiGGS Study: For the KiGGS baseline study, venous blood was collected after various (non-standard) fasting periods. TSH, fT3, and fT4 levels were measured using an electrochemiluminescence binding assay (Elecsys 2010, Roche Diagnostics, Mannheim, Germany). The interserial coefficients of variation over the entire study period were 3.9%, 5.9%, and 5.3% retrospectively [49]. For the KiGGS Wave 2 study, TSH, fT3, and fT4 levels were measured in the Central Epidemiological Laboratory of the Robert Koch Institute, Berlin, using an immunoassay on the Architect-Analyzer CI 8200 (Abbott, Abbott Park, North Chicago, IL, USA).

#### 4.1.5. Case Histories

Patients provided written informed consent for participation in the study and for the publication of patient images. Ethical approval for the study was obtained from the Institutional Review Board of Charité (EA2/026/20). The study was conducted in accordance with the tenets of the Declaration of Helsinki.

The male patient 1 (Figure 4, blue squares) with MCT8 deficiency was born at term after a normal pregnancy. During the first months of life, he was severely floppy, triggering a two-year diagnostic odyssey. Trio-exome sequencing revealed a hemizygous 1 bp insertion in *SLC16A2* (c.1235dupG; p.L413Pfs*25) causing a frameshift. The patient developed a severe phenotype of AHDS, with muscular hypotonia progressing to hypertonia of the extremities (spastic-dystonic) with persistent axial hypotonia, severe global developmental delay (unstable head control, inability to sit at 6 years of age), intellectual disability, dysphagia, and severe underweight requiring a permanent gastrostomy tube (Appendix A). Repeated measurements showed a highly elevated fT3/fT4 ratio.

The male patient 2 (Figure 4, blue diamonds) with MCT8 deficiency was born at term by Cesarean section after a normal pregnancy. During infancy, the patient showed mild hypotonia and developmental delay with head control at 10 months, free sitting at 12 months, and pulling to a standing position at 18 months. Later, he developed dystonia with “involuntary movements and pulling of the arms and legs” and hypersalivation (dysphagia) (Appendix A). Intellectual disability was suspected. Trio-exome sequencing revealed a novel *SLC16A2* missense mutation (c.1399G>A; p.G467S) that was not listed in gnomAD. In this mildly affected patient, fT3 was only slightly elevated.

The male patient 3 (Figure 4, blue triangles) MCT8 deficiency was born at term after a normal pregnancy. In the second year of life, he presented with mild developmental delay, with the ability to walk unaided at 17 months and to speak his first words at 19 months. At 6 years of age, he was also diagnosed with mild intellectual disability and intermittent “twisting/cramping of the hands” (suspected action-specific dystonia) (Appendix A). Trio-exome sequencing revealed a highly conserved *SLC16A2* missense variant (c.1378A>T; p.I460F) that was not listed in gnomAD. In this patient with a very mild phenotype of AHDS, the elevated fT3 was only slightly above the 97th percentile.

### 4.2. Statistical Methods

#### 4.2.1. Cleaning of Raw Data

Raw data were cleaned by removing individuals with a history of thyroid disease, TSH levels above 10 mU/L, medications affecting thyroid hormone metabolism (e.g., L-thyroxine, iodine), and increased iodine excretion above 1.08 mg/d.

#### 4.2.2. Generation the Centile Charts

The centile charts were generated using the R package ‘childsds’ (Data and Methods Around Reference Values in Pediatrics) v0.8.0 by Mandy Vogel [50]. The package allows centile charts to be generated from different cohort measurements; they are then stored as standards and individual SDS values are calculated from these standards. The distribution parameters were estimated assuming a BCPEo (Box–COX power exponential) distribution. The algorithm used the gamlss function from the ‘gamlss’ R package [51]. Linear regression plots were generated using the geom_smooth(method = lm) function of the ‘ggplot2’ R package [52].

## 5. Conclusions

Thyroid hormone resistance due to MCT8, THRα, and SECISBP2 defects may lead to a severe developmental delay like congenital hypothyroidism, but is characterized by normal TSH and partially normal T3 and T4 levels. This often delays diagnosis. Since there are therapeutic options, the diagnosis should be made as early as possible. Therefore, in the differential diagnosis of children with developmental delay, we recommend not only the determination of TSH as the standard parameter for normal thyroid function, but also the measurement of fT4 and fT3 levels to determine the fT3/fT4 ratio. While our data suggest that the fT3/fT4 ratio is pathognomonic from an early age, further studies are needed to establish the fT3/fT4 ratio as a possible neonatal screening parameter.

## Figures and Tables

**Figure 2 ijms-25-08585-f002:**
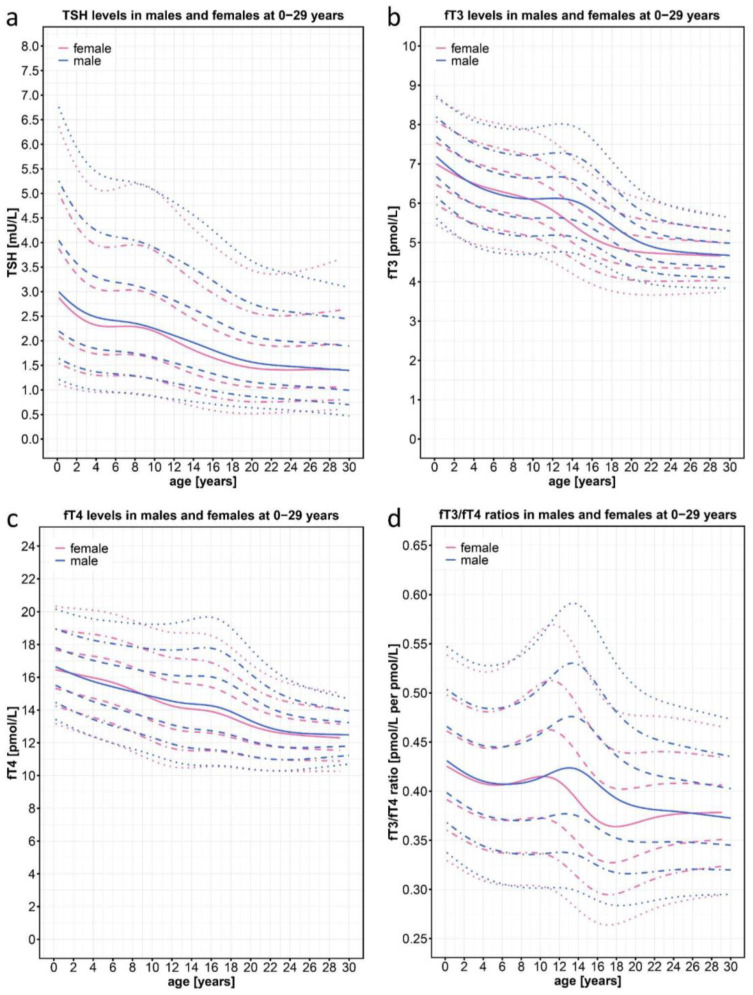
Normal percentiles of TSH, fT3, fT4, and the fT3/fT4 ratio derived from n = 23,522 data points of individuals (n = 11,325 females; n = 12,197 males) aged 0–29 years from the joint KiGGS baseline, KiGGS Wave 2, and LIFE studies based on the general pediatric and young adult population of Germany. The colored lines represent the 97th, 90th, 75th, 50th, 25th, 10th, and 3rd percentiles; pink represents female patients and blue represents male patients. NB: Point clouds for the fT3/fT4 centile charts are shown on Appendix A.

**Figure 3 ijms-25-08585-f003:**
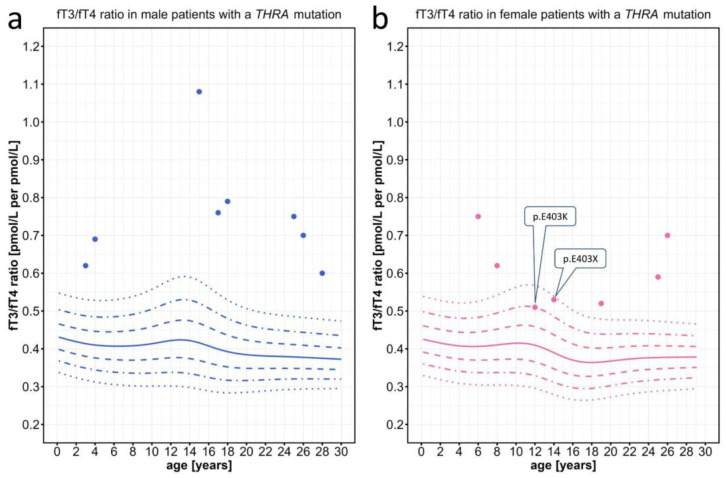
Individual fT3/fT4 ratio data points plotted against the percentiles for male and female patients with *THRA* mutations. Information on the individual patients can be found in Appendix A. The colored lines represent the 97th, 90th, 75th, 50th, 25th, 10th, and 3rd percentiles. The two female patients with the lowest fT3/fT4 ratio carried mutations in the C-terminal part of the THRα protein.

**Figure 4 ijms-25-08585-f004:**
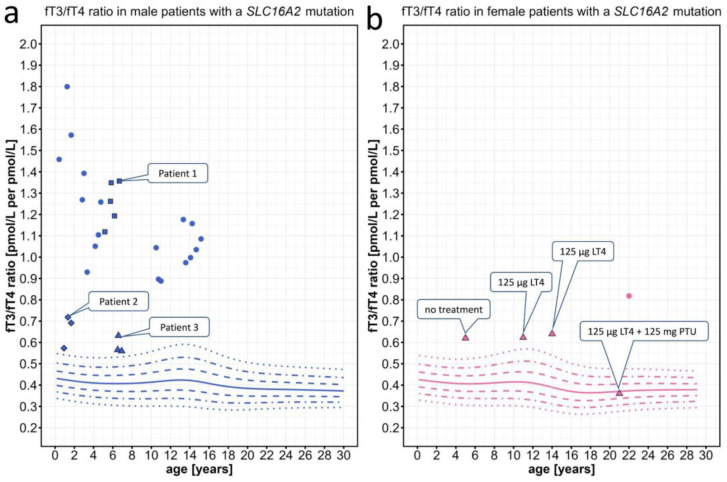
Individual fT3/fT4 ratio data points plotted against the percentiles for male and female patients with *SLC16A2* mutations. Information on individual patients can be found in Appendix A. (**a**) We present repeated measurements for two mildly affected male patients (diamonds, patient 2; triangle, patient 3) and one severely affected male patient (squares, patient 1). (**b**) Triangles represent a female patient treated with levothyroxine (LT4) and later by addition of 6-n-propyl-2 thiouracil (PTU), which normalized the fT3/fT4 ratio. The colored lines represent the 97th, 90th, 75th, 50th, 25th, 10th, and 3rd percentiles.

**Figure 5 ijms-25-08585-f005:**
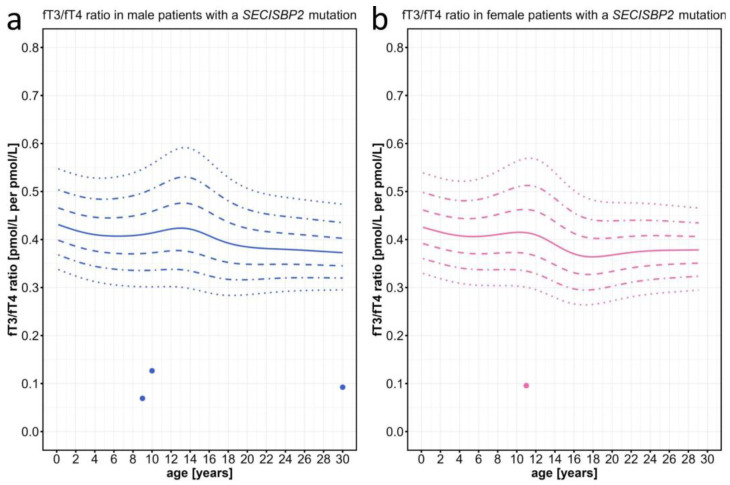
Individual fT3/fT4 ratio data points plotted against the percentiles for male and female patients with *SECISBP2* mutations. Information on individual patients can be found in Appendix A. The colored lines represent the 97th, 90th, 75th, 50th, 25th, 10th, and 3rd percentiles.

**Figure 6 ijms-25-08585-f006:**
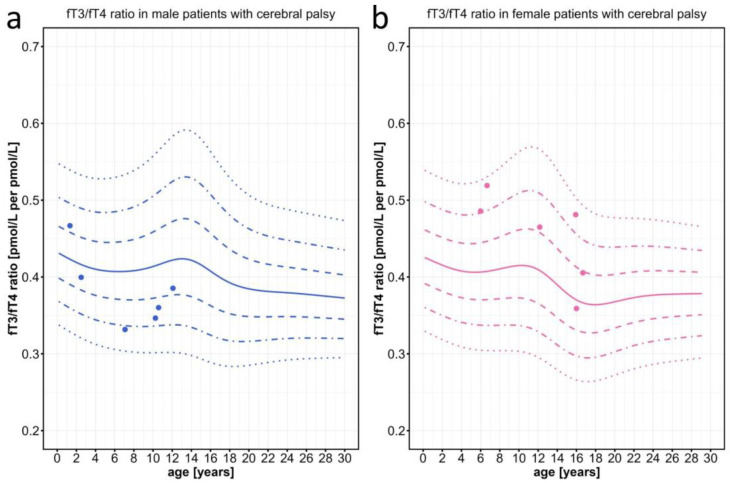
Individual fT3/fT4 ratio data points plotted against the percentiles for male and female patients with severe cerebral palsy, who served as pathological controls with global developmental delay and without any problems in their thyroid hormone system. Information on the patients can be found in Appendix A. The colored lines represent the 97th, 90th, 75th, 50th, 25th, 10th, and 3rd percentiles.

## Data Availability

KiGGS data provided by the Robert Koch Institute, Berlin: The authors confirm that some access restrictions apply to the data on which the results are based. The dataset cannot be made publicly available because the informed consent of the study participants did not cover public storage of the data. However, the minimal data set underlying the findings is archived at the Research Data Center of the Robert Koch Institute (RKI) and can be accessed by researchers upon reasonable request. On-site access to the data set is possible at the Secure Data Center of the RKI Research Data Center. Requests should be addressed to the Research Data Center, Robert Koch Institute, Berlin, Germany, at fdz@rki.de. Data from the LIFE Child Study, University of Leipzig: The data set presented in this article cannot be made publicly available due to ethical and legal restrictions. The LIFE Child Study is a study that collects potentially sensitive information. The publication of data is not covered by the informed consent of the study participants. In addition, LIFE’s privacy policy requires all researchers (both external and internal) interested in accessing data to sign a project agreement. Researchers interested in accessing data from the LIFE Child Study can contact the study by writing to forschungsdaten@medizin.uni-leipzig.de.

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
