# Peer review of "Normal Values for the fT3/fT4 Ratio: Centile Charts (0–29 Years) and Their Application for the Differential Diagnosis of Children with Developmental Delay"

_ijms, 2024, doi:10.3390/ijms25168585_

Round 1

Reviewer 1 Report

Comments and Suggestions for Authors

This is a large study aiming to generate normal values and sex-specific  percentiles for the ratio of free T3 to free T4 (fT3/fT4 ratio) with the objective to help the differential diagnosis of  the cases with thyroid hormone resistance (MCT8, THRα, and SECSIBP2 defects).

Indeed these were proven to have a clearly abnormal abnormal fT3/fT4 ratio.

The manuscript is the result of a large body of careful work, resulting in the generation of a readily available screening parameter to be assessed using a free online tool (fT3/fT4 ratio).

The references are appropriate, the data quality is exceptional, the conclusions are entirely sustained by the data presented.

In my opinion the work is extremely useful and interesting and definitely worth publishing.

Author Response

Reviewer #1

We thank the reviewer for his/her encouraging remarks and comments.

Reviewer 2 Report

Comments and Suggestions for Authors

Please see the attached review.

Comments on the Quality of English Language

Author Response

Reviewer #2

We thank the reviewer for his/her comments to which we will answer here below point-by-point..

  1. Title- should refer to thyroid hormone resistance.

Answer: We would like to keep the title as it is, because the article is not only aimed at endocrinologists but also at child neurologists who see these patients the first place. In most children with MCT8- and THRA-deficiency, "developmental delay" is the key symptom and presenting complaint. By our article we want to facilitate the discovery of children with peripheral thyroid hormone resistance amongst the large group of children with "developmental delay" and provide the necessary tools (e.g. web-page and reference values)

  1. Introduction- I would recommend to shorten some parts that include basic knowledge.

Answer: Also here, we would prefer to keep the introduction as it is, because we cannot make sure that all readers of the article are endocrinologists and thereby well versed in the intricacies of central and peripheral thyroid hormone action and control.

  1. 29: In contrast, in the rare disorders of thyroid hormone resistance, TSH and often also thyroid hormone levels are within the normal range.- it is not always the case, please discuss further.

Answer: We have now changed in the test that in affected children TSH and thyroid hormone levels may be in the normal range.

  1. 30: Thyroid hormone resistance is caused by defects in hormone metabolism, transport, or receptor activation and has the same severe consequences for childhood development as congenital hypothyroidism.- it is not always true, especially in the cases of THRA mutations, that may give milder clinical picture.

Answer: We have now discussed the phenotype of patients with THRA mutations who may have a milder phenotype

  1. 34: The aim was to determine whether cases with genetically confirmed thyroid hormone resistance (MCT8, THRα, and SECSIBP2 defects) had an abnormal fT3/fT4 ratio.- the Authors do not mention the THRbeta mutations at all.

Answer: The aim was to determine whether cases with developmental delay and genetically confirmed thyroid hormone resistance (MCT8, THRα, and SECSIBP2 defects) had an abnormal fT3/fT4 ratio. The article is not about patients with THRB mutations, because these patients generally do not present with developmental delay.

  1. 206: We first evaluated data from patients with THRA mutations, which had been extracted from the literature.- references are needed.

Answer: Originally we had provided the citations in Table A1 only as PMID numbers. We now provide the full citations in the footer of Table A1.

  1. 217: In patients with MCT8 deficiency and proven SLC16A2 mutations, the fT3/fT4 ratios were clearly located above the 97th percentile (Figure 4a).- as above.

Answer: We do not understand to what point the reviewer refers to exactly.

  1. 253: Finally, as a negative control, we tested the fT3/fT4 ratios of a cohort of children who were severely affected by cerebral palsy (CP) due to perinatal brain damage.- where have these children been treated? How many and at what age?

Answer: The n = 6 male and n = 6 female children are presently being treated at our Center for Chronically Sick Children (Charité). The actual age of the children can be derived from the X-axis of Figure 6.

  1. 265-282- information repeated

Answer: As a question of style we would prefer to repeat some key points at the beginning of the discussion and would therefore prefer to leave this passage unchanged.

  1. 299- the treatment options should be discussed further.

Answer: We have now cited a paper [Reference 30] detailing the treatment options for patients with THRA mutation (with L-thyroxine).

  1. 379- what was the treatment in the patients with MCT8-deficiency (additionally this is a preferred form rather than MCT8-deficient patients).

Answer: We have now replaced all occurrences of "MCT8-deficient patients" by "patients with MCT8-deficiency". The treatment of one female patient has been mentioned in Table A1 and in the Legend of Figure 4. All other patients were only treated symptomatically for their neurological symptoms (e.g. BoTox). None of the patients had received medication with an influence on the thyroid hormone levels.

  1. 420: Thyroid hormone resistance due to MCT8, THRα, and SECISBP2 defects leads to severe developmental delay like congenital hypothyroidism but is characterized by normal TSH and partially normal T3 and T4 levels.- please see the above comments.

Answer: We changed the text to: "Thyroid hormone resistance due to MCT8, THRα, and SECISBP2 defects may lead to severe developmental delay like congenital hypothyroidism but is characterized by normal TSH and partially normal T3 and T4 levels".

  1. Lack of important ref regarding THRA mutations: J Med Genet 2015;0:1–5. doi:10.1136/jmedgenet-2014-102936

Answer: The paper had been cited in the supplementary Table A1. Now we have also cited it in the main body of the manuscript.

Round 2

Reviewer 2 Report

Comments and Suggestions for Authors

Thank you for the answer. However I cannot find the new version of the manuscript with “truck changes”.

1.     In my opinion Title does not exactly refer to the text. Moreover I would still strongly recommend to improve Introduction and avoid repeating information as pointed in my first review.

2.     THRbeta mutations should be mentioned and explained why they are not within the aim of the study.

3.     206: We first evaluated data from patients with THRA mutations, which had been extracted from the literature.- references are needed in the main text.

4.     217: In patients with MCT8 deficiency and proven SLC16A2 mutations, the fT3/fT4 ratios were clearly located above the 97th percentile (Figure 4a).- as above, references are needed in the main text.

5.     253: Finally, as a negative control, we tested the fT3/fT4 ratios of a cohort of children who were severely affected by cerebral palsy (CP) due to perinatal brain damage.- where have these children been treated? How many and at what age?- this data should be presented in the main text.

6.     299- the treatment options should be discussed in more detail.

7.     379- the treatment in the patients with MCT8-deficiency should be presented in the main text.

Comments on the Quality of English Language

Minor editing of English language required

Author Response

We thank reviewer #2 for additional comments. Please find a point-to-point answer below.

Reviewer #2

Thank you for the answer. However I cannot find the new version of the manuscript with “truck changes”.

Answer: As there was no possibility at the journal submission site to separately upload a track-change version, we had included this into the supplementary materials. We now provide a track-change as a comparison between this second revision and the original submission. Please find it within the folder of the supplementary materials.

  1. In my opinion Title does not exactly refer to the text. Moreover I would still strongly recommend to improve Introduction and avoid repeating information as pointed in my first review.

Answer: As in our previous answer, the main aim of the article is to provide reference values for the fT3/fT4 ratio in different age groups. Secondly, we want to show that these reference values are useful to filter out children with thyroid hormone resistance syndromes from the large group of children with developmental delay, who are primarily seen by child neurologists and not by endocrinologists. We would like to keep this clinical focus towards child neurologists and hence would like to keep the introduction a bit more basic and extensive as would be needed for endocrinologists.

  1. THRbeta mutations should be mentioned and explained why they are not within the aim of the study.

Answer: We refer to the answer in our previous rebuttal. This is not a review about thyroid resistance syndromes in general. Our starting point are children with developmental disorders, who present to child neurologists, and whose TSH levels are normal and who often have fT3 and fT4 levels (still) within the normal range. Children with mutations in TSHB have neither normal TSH levels nor do they suffer from developmental delay and are therefore not subject of this article. In the manuscript we have now added the following line: "Patients with a THRβ defect have impaired thyroid hormone sensing by the pituitary, resulting in elevated TSH and consequently elevated T4 and T3, by which the diagnosis can be easily made. Clinical symptoms of these children may result from overstimulation of the unaffected THRα by the elevated plasma hormones leading to hyperthyroid symptoms [13,14]."

  1. 206: We first evaluated data from patients with THRA mutations, which had been extracted from the literature.- references are needed in the main text.

Answer: We have now cited all the paper references from which we had cited laboratory values in Table A1 also in the main body of the manuscript (in addition to Table A1).

  1. 217: In patients with MCT8 deficiency and proven SLC16A2 mutations, the fT3/fT4 ratios were clearly located above the 97th percentile (Figure 4a).- as above, references are needed in the main text.

Answer: Since all the patients with MCT8-deficiency were attended by one of the (co)authors of this manuscript or physicians mentioned in the acknowledgment, we only used our own lab values. Therefore, we do not need to cite references here.

  1. 253: Finally, as a negative control, we tested the fT3/fT4 ratios of a cohort of children who were severely affected by cerebral palsy (CP) due to perinatal brain damage.- where have these children been treated? How many and at what age?- this data should be presented in the main text.

Answer: We have now added the required information into the respective section: "Finally, as a negative control, we tested the fT3/fT4 ratio in a cohort of n = 12 children between 1.3-16.7 years of age with severe cerebral palsy (CP) due to perinatal brain damage, who had come to our outpatient department for treatment with botulinum toxin injections."

  1. 299- the treatment options should be discussed in more detail.

Answer: At line 299 of the manuscript no treatment is mentioned. Beyond that, a discussion of the treatment options for children with thyroid hormone resistance would go well beyond the scope of this article, which deals with diagnostics, not with treatment. Nevertheless we have now mentioned the treatment of THRA mutations with L-thyroxine: "This is especially true due to the therapeutic options that are available, such as the treatment of patients with THRA mutations with L-thyroxine [39] and that will be developed in the near future for these thyroid hormone resistance diseases." We do not know of any other specific treatment approaches at the moment.

  1. 379- the treatment in the patients with MCT8-deficiency should be presented in the main text.

Answer: Presently no causative treatment for children with MCT8-deficiency beyond supportive treatment such as physiotherapy is available. Neither TRIAC I nor TRIAC II study substantially improved the neurological outcome of the children.

Comments on the Quality of English Language. Minor editing of English language required

Answer: The text has been edited by a native speaker.
